# SMAD4 Expression in Monocytes as a Potential Biomarker for Atherosclerosis Risk in Patients with Obstructive Sleep Apnea

**DOI:** 10.3390/ijms24097900

**Published:** 2023-04-26

**Authors:** Elena Díaz-García, Aldara García-Sánchez, David Sánz-Rubio, Enrique Alfaro, Cristina López-Fernández, Raquel Casitas, Eva Mañas Baena, Irene Cano-Pumarega, Pablo Cubero, Marta Marin-Oto, Eduardo López-Collazo, José María Marin, Francisco García-Río, Carolina Cubillos-Zapata

**Affiliations:** 1Biomedical Research Networking Centre on Respiratory Diseases (CIBERES), 28029 Madrid, Spain; 2Respiratory Diseases Group, Respiratory Diseases Department, La Paz University Hospital, IdiPAZ, 28046 Madrid, Spain; 3Servicio de Neumología, Hospital Universitario Ramón y Cajal, 28034 Madrid, Spain; 4Precision Medicine in Respiratory Diseases Group, Miguel Servet University Hospital-IIS Aragon, 50009 Zaragoza, Spain; 5The Innate Immune Response Group, La Paz University Hospital, IdiPAZ, 28046 Madrid, Spain; 6Department of Medicine, University of Zaragoza School of Medicine, 50009 Zaragoza, Spain; 7Faculty of Medicine, Autonomous University of Madrid, 28029 Madrid, Spain

**Keywords:** SMAD4, OSA, NLRP3, TGF-β, inflammation, TF, atherosclerosis, hypoxia, HIF1α

## Abstract

Obstructive sleep apnea (OSA) patients are at special risk of suffering atherosclerosis, leading to major cardiovascular diseases. Notably, the transforming growth factor (TGF-β) plays a crucial role in the development and progression of atherosclerosis. In this context, the central regulator of TGF-β pathway, SMAD4 (small mother against decapentaplegic homolog 4), has been previously reported to be augmented in OSA patients, which levels were even higher in patients with concomitant cardiometabolic diseases. Here, we analyzed soluble and intracellular SMAD4 levels in plasma and monocytes from OSA patients and non-apneic subjects, with or without early subclinical atherosclerosis (eSA). In addition, we used in vitro and ex vivo models to explore the mechanisms underlying SMAD4 upregulation and release. Our study confirmed elevated sSMAD4 levels in OSA patients and identified that its levels were even higher in those OSA patients with eSA. Moreover, we demonstrated that SMAD4 is overexpressed in OSA monocytes and that intermittent hypoxia contributes to SMAD4 upregulation and release in a process mediated by NLRP3. In conclusion, this study highlights the potential role of sSMAD4 as a biomarker for atherosclerosis risk in OSA patients and provides new insights into the mechanisms underlying its upregulation and release to the extracellular space.

## 1. Introduction

Obstructive sleep apnea (OSA) is a very prevalent disorder characterized by recurrent collapses of the upper airways causing intermittent hypoxia (IH) and sleep fragmentation. These alterations are known to induce oxidative stress, systemic inflammation, sympathetic activation, and metabolic deregulation [1], providing substrate conditions for the development of cardiovascular and metabolic disease [2,3,4]. In fact, OSA is associated with substantial comorbidities, including cardiovascular and metabolic diseases [4,5,6], constituting the cornerstone of OSA-related excess mortality [7]. In this line, atherosclerosis plays a substantial role in the interplay between OSA and cardiovascular morbidity [8], facilitating the development of major complications such as ischemic heart disease, cerebrovascular events, arrhythmias, or heart dysfunction [9]. Interestingly, OSA patients are characterized by a pro-inflammatory phenotype and chronic inflammation is thought to be responsible for atherosclerosis progression. Thus, the activation of inflammatory responses has been suggested to play an important role in the association between OSA and atherosclerosis [10]. Indeed, NLRP3 (nucleotide-binding oligomerization domain-like receptor protein 3) inflammasome has been identified as a key pro-inflammatory mechanism in OSA patients [11]. In particular, NLRP3 activation in OSA patients has been shown to be mediated by the IH and the presence of high levels of oxidized low-density lipoprotein [12]. Interestingly, NLRP3 activation has been found to be especially strong in OSA patients with early subclinical atherosclerosis (eSA) providing a possible mechanism for OSA-associated atherosclerotic risk [12]. In agreement, NLRP3 activation in OSA patients has been shown to induce the release of the coagulation initiator tissue factor (TF), that promotes systemic coagulation, thrombosis, and atherosclerosis progression [13,14]. These findings suggest a role of the NLRP3–pyroptosis axis in the release of TF into plasma [15], which could also be involved in the release to the plasma of other intracellular mediators.

Besides, other studies focusing on inflammation have suggested that activation of transforming growth factor β (TGF-β) signaling may be an important driver of atherosclerosis-associated inflammation [16,17]. These findings are difficult to reconcile with a general perception of the anti-inflammatory effects of TGF-β signaling. Thus, TGF-β role in atherosclerosis has been controversial with both pro- and anti-atherosclerotic effects reported [18,19]. Indeed, the TGF-β/Smad pathway’s pro-atherogenic role is mediated by its ability to promote fibrosis and inhibit endothelial regeneration [20,21]. On the other hand, the TGF-β pathway could also limit atherosclerosis progression by modulating the inflammatory response and preventing lipid accumulation in the vessel wall [22]. Interestingly, OSA patients frequently present an increase in TGF-β levels [15,23,24] suggesting that the pathways dependent on its activation could play a role in the high rates of atherosclerotic patients among OSA subjects.

In this context, Smad (small mother against decapentaplegic homolog) is the main proteins driving canonical TGFβ signaling [25]. Briefly, TGF-β receptors phosphorylate receptor (R-) Smads (Smad2 and Smad3) [25,26]; then phosphorylated R-Smads oligomerize with the common mediator (Co-) Smad4; finally, this Smad4 complex translocate to the nucleus and regulates gene expression. Importantly, Smad4 is the unique co-Smad described to date [27]. In this line, various studies show that Smad4 inhibition suppresses TGF-β signaling [28,29,30]. Hence, Smad4 is considered the central mediator of the canonical TGF-β pathway, contributing to the pleiotropic functions of this cytokine in both physiological and pathological processes [31]. Indeed, gain-of-function mutations in SMAD4 lead to a plethora of cardiovascular abnormalities such as congenital heart defects, restrictive cardiomyopathy, and systemic hypertension [32]. Conversely, other studies suggest that Smad4 could exert a protective role, since its inhibition enhanced inflammation, reversed cholesterol transport, and promoted atherosclerosis by coordinating regulation of several genes related to the pathogenesis of this disorder [33]. Thus, Smad4 could act as a complex and multifaceted regulator on metabolic homeostasis, fibrosis, matrix remodeling, and inflammation [34]. Interestingly, elevated soluble SMAD4 (sSMAD4) protein has been found in the plasma of OSA patients, the levels of which were even higher in those subjects with cardiometabolic comorbidities such as dyslipidemia or hypertension [35], two major risk factors for the development of atherosclerosis [36]. All this evidence prompted us to speculate that Smad4 could play a role as a potential biomarker for atherosclerosis risk in OSA patients. Additionally, since plasmatic Smad4 origin has not been explored, we hypothesized that sSMAD4 is released from OSA monocytes in a NLRP3-dependent manner, as it occurs with TF. Therefore, in this study, we analyzed the expression of Smad4 in OSA patients with or without early subclinical atherosclerosis and assessed potential mechanisms driving its overexpression and release to the plasma.

## 2. Results

### 2.1. Subject Characteristics

The main characteristics of the subjects from the EPIOSA cohort are summarized in Table 1. In particular, in the EPIOSA cohort, as expected, the presence of OSA or eSA was associated with older age and higher BMI and blood pressure, with no differences in sex, cumulative tobacco use, or sleepiness.

On the other hand, the Ramon y Cajal (RC) cohort includes patients with severe OSA and healthy subjects, homogeneous in terms of sex, age, BMI, and smoking habit (Table 2).

### 2.2. Patients with Atherosclerosis Exhibited Elevated sSMAD4 Levels

As a first approach, we assessed the possible association of sSMAD4 plasmatic levels with the presence of atherosclerotic plaques. For this purpose, we used the EPIOSA cohort because clinical parameters related to atherosclerosis were available (carotid intima-media thickness [IMT]). Interestingly, sSMAD4 levels were higher in patients with atherosclerotic plaques (Figure 1A) and correlated with intima-media thickness (Figure 1B). Moreover, sSMAD4 levels were also related to the pro-coagulant marker TF (Figure 1C), suggesting a possible role of sSMAD4 as a novel biomarker for cardiovascular risk.

### 2.3. sSMAD4 Levels Are Elevated in Patients with OSA and eSA and Related to OSA Severity

In order to explore the possible role of sSMAD4 as an early biomarker for atherosclerosis, we analyzed its levels in EPIOSA patients with early subclinical atherosclerosis (eSA) without coexistent atherosclerotic plaques. Our results indicated that sSMAD4 levels were higher in OSA patients with and without eSA when compared with HS without eSA. Importantly, sSMAD4 levels were higher in OSA-eSA patients than in OSA-non-eSA (Figure 2A). Differences between HS-non-eSA and OSA groups remained after adjusting for age, sex, BMI, and blood pressure (Figure 2B). Further, sSMAD4 plasma levels positively correlated with AHI, oxygen desaturation index (ODI), time with oxygen saturation less than 90% (CT90), and mean and minimum oxygen saturation (SaO_2_) (Figure 2C,D and Appendix A). Accordingly, patients with severe OSA exhibit higher levels of sSMAD4 than mild OSA patients (Appendix A).

Moreover, SMAD4 mRNA levels were also higher in the peripheral blood mononuclear cells (PBMCs) of patients with OSA and correlated with disease severity parameters (Appendix A, Appendix A). This evidence suggests sSMAD4 levels are upregulated under OSA conditions and increase along OSA severity.

### 2.4. Intracellular SMAD4 Levels Where Higher in Monocytes from Patients with OSA 

Although high levels of plasmatic sSMAD4 in OSA patients have already been reported [35], the mechanism underlying the release of this protein to plasma is currently unknown. In this context, we decided to explore the possible origin of plasmatic sSMAD4. According to our previous data, we found that sSMAD4 was increased in patients with OSA and correlated with AHI, ODI, mean, and minimum SaO_2_ (Figure 3A–C and Appendix A). Remarkably, sSMAD4 levels were in the same range as in the EPIOSA cohort. In addition, our results show that intracellular expression of SMAD4 was increased in monocytes from patients with OSA (Figure 3D). Moreover, intracellular SMAD4 levels also correlated with AHI, ODI, mean, and minimum SaO_2_ (Figure 3E,F and Appendix A). Furthermore, SMAD4 mRNA expression was higher in PBMCs from patients with severe OSA than in healthy subjects, and SMAD4 mRNA levels correlated with sleep parameters (Appendix A, Appendix A). Collectively, our results suggest that OSA increases SMAD4 expression in monocytes, which could be a potential source contributing to the high levels of plasmatic SMAD4.

### 2.5. SMAD4 Expression and Release Is Related to Intermittent Hypoxia

We then explored the possible mechanisms underlying SMAD4 overexpression in OSA patients. In this context, we created an in vitro model of IH using monocytes from healthy volunteers as we have previously described [11,35]. Interestingly, monocytes incubated under IH exhibited higher levels of SMAD4, both at the protein and mRNA levels (Appendix A). Furthermore, IH also increases SMAD4 release to the plasma (Appendix A), suggesting that IH mediates both overexpression and release of SMAD4 to the extracellular space. Additionally, we treated healthy volunteers monocytes with PX-478 (S-2-amino-3-[4′-N,N,-bis(2-chloroethyl)amino]phenyl propionic acid N-oxide dihydrochloride) an inhibitor of HIF1α expression and activity [37,38,39]. Our data showed that SMAD4 release was decreased in PX478-treated cells (Appendix A), indicating that IH, through HIF1α, mediates SMAD4 overexpression and release to the plasma.

### 2.6. NLRP3 Inflammasome Could Mediate SMAD4 Release to the Plasma

As previously published, our results showed an upregulation of NLRP3 in monocytes from patients with severe OSA (Appendix A). Moreover, sSMAD4 correlated with TF plasma levels (Figure 4A) and NLRP3 correlated with SMAD4 intracellular levels (Figure 4B), indicating a possible role of NLRP3 inflammasome in SMAD4 release to plasma. Thus, to assess if NLRP3 inflammasome might trigger the release of SMAD4 to the extracellular space, we perform an ex vivo model using monocytes from healthy subjects and OSA patients treated or not with two pharmacological inhibitors targeting both NLRP3 (MCC-950) and caspase-1 (Ac-YVAD-cmk), then the supernatant protein content was analyzed (Figure 4C). Our results showed a significant increase in supernatant SMAD4 in OSA monocytes compared with HS cultures. Moreover, our data suggested a decrease of sSMAD4 levels in OSA monocyte supernatant after both treatments (Figure 4D). Collectively, these data suggest that NLRP3 inhibition (mainly expressed in monocytes/macrophages) leads to a decrease in SMAD4 release, indicating that this population has a major contribution to SMAD4 release to the supernatant. Nevertheless, the assays were performed with enriched monocytes culture; thus, we could not dismiss that other PBMC populations could be contributing to sSMAD4 concentration in the supernatant. Taken together, these data indicate that severe OSA through NLRP3 upregulates sSMAD4 release into the plasma.

## 3. Discussion

Our study confirmed that sSMAD4 is increased in plasma from patients with OSA and, remarkably, this protein that is associated with atherosclerosis progression is higher in those patients with OSA and eSA. Furthermore, in vitro and ex vivo assays elucidated a possible mechanism of releasing this protein to the plasma from monocytes through the NLRP3 inflammasome. Those findings suggest the pivotal role of SMAD4 in the development of eSA associated with OSA.

OSA patients are at special risk of suffering atherosclerosis since they exhibit lipid dysregulation in combination with a low-grade inflammatory basal condition [4,11,12,19,40,41]. Importantly, sustained low-grade inflammation together with high circulating cholesterol levels has widely been associated with the development of atherosclerosis [42,43]. In this context, a plethora of studies showed that TGFβ1/Smad signaling pathway played a central role in cardiovascular and cerebrovascular diseases, including atherosclerosis [16,17,44,45,46,47]. Interestingly, it is reported that the TGF-β pathway is upregulated in OSA patients, and its activity is related to the hypoxemia severity [23,24,48,49,50,51]. Although TGF-β has been classically defined as an anti-inflammatory cytokine, its role in inflammation has been recently reviewed [27,52]. In particular, the TGF-β signaling signature and expression of TGF-β ligands, receptors, and various Smad proteins have been reported in atherosclerotic plaques [47]. In fact, activation of TGF-β signaling has been shown to play a key role in the induction of vessel wall inflammation and the development and progression of atherosclerosis [16]. In this context, Smad4 plays a central role as a crucial mediator driving the TGF-β signaling pathway from the cell membrane to the nucleus [53]. In fact, various studies show that suppression of Smad4 disrupts canonical TGF-β signaling [28,29,30]. We previously reported sSMAD4 overexpression in OSA patients [35], and here we corroborated this finding data by analyzing additional cohorts. Here, our data from both cohorts included in this study were in accordance with previous results from our lab, identifying high levels of sSMAD4 in patients with OSA, especially in those with cardiometabolic comorbidities [35]. More importantly, the current study identifies that sSMAD4 is especially upregulated in OSA patients with eSA, highlighting its possible role as a biomarker for atherosclerotic risk. 

Since the Smad family has been typically defined as comprising intracellular signal transductor proteins, a major concern of this study was to elucidate the origin of plasmatic sSMAD4. Interestingly, previous studies reported elevated levels of Smad proteins in the serum of patients with major diseases of pleiotropic origin such as non-small cell lung cancer or coronary artery disease [54,55]. However, none of these studies assessed the source of these Smad soluble forms, nor explored mechanisms underlying Smad delivery to the extracellular space. In this context, our study brings out the identification of high levels of intracellular SMAD4 in monocytes from OSA patients. Moreover, we found that IH could be underlying SMAD4 upregulation in OSA patients, since in vitro exposure to IH is capable of inducing SMAD4 upregulation in healthy monocytes, contributing to its release to the extracellular space in a HIF-1α dependent manner. These results are in line with previous studies reporting an upregulation of the TGF-β/Smad4 pathway upon IH conditions both in in vitro and animal models [23,35,56]. Taken together, this data suggests monocytes as the potential source of sSMAD4 in the plasma. Here, we focused our research on monocytes/macrophages since they are well established as main players in the development of atherosclerosis due to their ability to drive cholesterol uptake and deposition and their contribution to inflammation [57]. However, it is predictable that monocytes are not the only source of sSMAD4 levels, and other cell types might be involved in sSMAD4 upregulation in OSA patients.

In this line, previous research from our group identified an upregulation of the NLRP3–pyroptosis axis in OSA patients with eSA from the EPIOSA cohort. Herein, our data validated that OSA monocytes released higher levels of sSMAD4 in comparison with healthy subjects in a process mediated by the activation of NLRP3. Besides, there is extensive experimental and clinical evidence confirming the central role of sterile inflammation in atherosclerosis [58]. Furthermore, NLRP3 has recently arisen as a central regulator of atherosclerosis initiation and progression [59]. In agreement, NLRP3 inflammasome has been reported to drive an increase in the plasmatic TF levels in OSA patients. Interestingly, we show that TF levels correlate with sSMAD4. Accordingly, previous data from our lab showed that TF correlates with OSA severity and is also higher in patients with cardiometabolic comorbidities including atherosclerosis [11,12]. Indeed, the release of TF to plasma has a functional impact due to its central role as a primary initiator of the coagulation cascade, thereby contributing to systemic coagulation, thrombosis, and atherosclerosis progression [13,14]. Conversely, whether Smad4 fulfills any functional role in the extracellular space remains completely unknown. Interestingly, other intracellular proteins with defined intracellular function are released outside the cell following tissue injuries such as HMGB1, histones, or cyclophilin A, initiating the immune response [60]. These proteins are normally located in the intracellular space, and upon their release, are detected by immune-related receptors such as TLRs and NLRs [61]. In fact, it has been reported that a variety of TLRs are markedly augmented in endothelial cells of human atherosclerotic lesions and cultured human vascular endothelial cells stimulated with pro-inflammatory cytokines [62,63,64]. In turn, it has been reported that activation of some TLRs such as TLR2 and TLR4, leads to the transcription of adhesion molecules such as VCAM and ICAM in a process mediated by NF-κB, facilitating atherosclerosis progression [65]. Therefore, sSMAD4 could potentially contribute to enhance the inflammatory response, thus contributing to atherosclerosis initiation or progression. Nevertheless, beyond mere speculation additional work is needed to elucidate the potential role of sSMAD4 as an extracellular protein.

Our study has several limitations, which we recognize. Firstly, we used two different OSA cohorts inducing potential confusional factors; however, both cohorts bring out similar results related to sSMAD4 overexpression in plasma from OSA patients. Second, PBMCs from EPIOSA cohort subjects were not available to assess SMAD4 monocyte protein expression or to perform ex vivo assays. Third, in the RC cohort carotid intima-media thickness (IMT) could not be assessed, so we could only analyze HS and OSA groups in this cohort. Fourth, we did not assess the potential function of sSMAD4 as an extracellular protein. Fifth, it was a cross-sectional analysis, so information about the longitudinal evolution of these patients is necessary to infer causality.

Collectively, our results showed that sSMAD4 levels were elevated in the plasma of patients with OSA and early subclinical atherosclerosis without further comorbid conditions. Importantly, we provided evidence suggesting that plasmatic sSMAD4 is released from monocytes in a process mediated by IH and NLRP3 inflammasome activation.

Overall, this study highlights the potential role of sSMAD4 as a biomarker for early atherosclerosis risk in OSA patients and opens the path for new research on the potential role of SMAD4 as a relevant player in cardiovascular disease. Finally, our data contribute to identifying new tools for the early identification of atherosclerosis risk in OSA patients, contributing to the development of personalized medicine in the therapeutic management of this disease.

## 4. Materials and Methods

### 4.1. Study Subjects

This study included two independent OSA cohorts. A detailed description of the selection criteria for both cohorts is provided online in Appendix A. In particular, the Epigenetics Status and Subclinical Atherosclerosis in Obstructive Sleep Apnea (EPIOSA). EPIOSA is a longitudinal study to assess potential epigenetic markers associated with the prevalence and progression of sub-clinical atherosclerosis in individuals with OSA without co-morbid conditions (Clinical-Trials.gov: NCT02131610) [26]. EPIOSA cohort recruited consecutive subjects aged 20 to 60 years old and free of any acute or chronic comorbid condition other than OSA. Briefly, after a home sleep test, subjects were allocated as OSA (apnea–hypopnea index [AHI] ≥ 10 events/hour of recording time) or healthy (AHI < 5). Common carotid intima-media thickness (IMT) was assessed using the Philips IU22 ultrasound system (Philips Healthcare, Bothell, WA, USA). Early subclinical atherosclerosis was defined when IMT was greater than the upper limit of normal of our local healthy population without the presence of carotid plaques [27]. Finally, the study groups comprised OSA patients (OSA) and healthy subjects (HS), with or without evidence of early subclinical atherosclerosis (non-eSA and eSA, respectively). From this cohort, plasma and buffy coat samples were available. 

Due to the non-availability of PBMC samples for flow cytometry analysis in the EPIOSA cohort, we recruited a new cohort to validate our data and to perform ex vivo models. Thus, 50 newly diagnosed severe OSA patients were consecutively recruited from the Pneumology Service of Ramón y Cajal University Hospital, Madrid, Spain (RC). The diagnosis of OSA was established by respiratory polygraphy (Embletta GOLD, ResMed, Madrid, Spain), which included a continuous recording of oronasal flow and pressure, heart rate, thoracic and abdominal respiratory movements, and oxygen saturation (SaO_2_). Those tests in which the patients claimed to sleep less than 4 h or in which there were less than 5 h of nocturnal recording were repeated. Patients were considered to have severe OSA when their AHI was greater than 30. In addition, healthy subjects (HS) were selected from the census register of Madrid, Spain, metropolitan area, paired for sex, age, and body mass index (BMI) with OSA patients. In all of the healthy subjects, the diagnosis of OSA was ruled out by respiratory polygraphy. From this cohort, plasma, and PBMCs were available as biological material. Unfortunately, carotid intima-media thickness (IMT) could not be assessed in this cohort so the presence of atherosclerotic plaques or early subclinical atherosclerosis could not be defined. 

All participants from both cohorts provided written informed consent and the study was approved by the local ethics committees. A flow chart summarizing included participants is displayed in Appendix A.

### 4.2. Plasma Protein Determination 

Plasma samples from all subjects from both cohorts were analyzed for sSMAD4 (CSB-E12749h from CUSABIO, Wuhan, China) and TF (CSB-07913h from CUSABIO, Houston, TX, USA) concentration using the respective human enzyme-linked immunosorbent assay (ELISA). We followed the manufacturer’s instructions for all samples. Measurements for plasma samples were carried out in duplicate. The detection limit of the assay was 25 pg/mL for SMAD4 and 0.312 pg/mL for TF. The intra-assay variation was below 20% in the two different assays.

### 4.3. Human Cell Isolation

PBMCs were isolated from RC cohort subjects by centrifugation using Ficoll-Paque Plus (Amersham Bioscience, Uppsala, Sweden) density gradient. The cells were cultured in Roswell Park Memorial Institute (RPMI) 1640 medium supplemented with 100 U/mL penicillin and 100 μg/mL streptomycin. The monocytes were enriched by adherence for 1 h in media culture without fetal bovine serum, as we have previously described [24,28]. We seeded 0.5 × 10^6^ monocytes per well (6-well plates). The medium was then replaced with fresh culture media supplemented with 10% fetal bovine serum. Cells were incubated for 16 h at 37 °C and 5% CO_2_.

### 4.4. Flow Cytometry

After 16 h, monocytes from RC patients were harvested and washed with PBS (phosphate-buffered saline). Then, cells were stained with anti-human CD14 (BV510) (BD Biosciences, Aalst, Belgium), anti-human SMAD4 (AF488) (R&D Systems, Minneapolis, MN, USA), and anti-human NLRP3 (phycoerythrin) (Miltenyi Biotec, Bergisch Gladbach, Germany), following a standard protocol for intracellular staining (Transcription Factor Buffer Set, BD Biosciences, Aalst, Belgium). Appropriate isotype controls were used for each experiment. After staining for 30 min at 4 °C in the dark, cells were washed with PBS. Finally, cells were acquired using BD FACS-Celesta flow cytometer (Becton-Dickinson Biosciences, Aalst, Belgium), and data were analyzed using FlowJo vX.0.7 software (FlowJo, Ashland, OR, USA). The gating strategy is shown in Appendix A.

### 4.5. mRNA Isolation and Quantification by qPCR

In samples from the EPIOSA cohort RNA was extracted from buffy coats from randomly selected HS-non-eSA (n = 9), HS-eSA (n = 8), OSA-non-eSA (n = 10), and OSA-eSA (n = 9) subjects using TRIzol™ reagent (Invitrogen, Waltham, MA, USA) following the manufacturer’s protocol. Meanwhile, in samples from the RC cohort RNA was obtained from PBMCs using the High Pure RNA Isolation Kit (Roche Diagnostics, Rotkreuz, Switzerland). In both cohorts, RNA was quantified and complementary DNA (cDNA) was obtained by reverse transcription of 1 μg RNA using the High-Capacity cDNA Reverse Transcription kit (Applied Biosystems, Waltham, MA, USA). cDNA levels were measured using CFX96 Touch Real-Time PCR Detection System (Bio-Rad Laboratories, Hercules, CA, USA), NZYSupreme qPCR Green Master Mix (2×) (NZYTech, Lisbon, Portugal), and specific SMAD4 primers 5′-TGCATTCCAGCCTCCCATTT-3′ (forward) and 5′-CTCTCCTACCTGAACGTC-CATT-3′ (reverse); and 18S primers 5′-CGGCGACGACCCATTCGAAC-3′ (forward) and 5′-GAATCGAACCCTGATTCCCCGTC-3′ (reverse). The results were normalized to the expression of 18S, and the cDNA copy number of each gene of interest was determined using a 6-point standard curve.

### 4.6. Intermittent Hypoxia In Vitro Model

For intermittent hypoxia (IH) conditions, healthy monocytes were cultured in an incubation chamber attached to an external oxygen/nitrogen computer-driven controller using BioSpherix OxyCycler C42 (Redfield, NY, USA), a system that generates periodic changes in oxygen concentrations and controls air gas levels in each chamber, while individually maintaining CO_2_ as previously described [24,29]. Our IH model cycled oxygen in the medium at 1% for 2 min, followed by 20% for 10 min, with CO_2_ maintained at 5%. Hypoxia-inducible factor (HIF1α) inhibition was performed by treating the monocytes with 30 μM PX-478 (MedKoo Biosciences, Morrisville, NC, USA) [30] for 16 h under routine culture conditions or IH conditions.

### 4.7. NLRP3 Inhibition Ex Vivo Model

NLRP3 inhibition was performed using monocytes from OSA or HS subjects, 0.5 × 10^6^ monocytes per well cultured in M6 plates and treated or not with 5 μM MCC-950 (INH-MCC, IbianTechnologies S.L., Zaragoza, Spain) or 30 μg/mL Ac-YVAD-cmk (SML0429, Merck Life Science, Darmstadt, Germany) for 16 h. Finally, supernatants were harvested for sSMAD4 concentration analysis by ELISA (CSB-E12749h from CUSABIO, Wuhan, China).

### 4.8. Statistical Analysis

Comparisons between two groups were performed using *t*-Student test, Wilcoxon test, or Mann–Whitney U test, according to the distribution of variables. Comparisons between more than two groups were performed using chi-squared, Kruskal–Wallis test, or Two-way ANOVA with post-hoc comparisons by Bonferroni test, according to the type and distribution of variables. Between-group comparisons of sSMAD4 were adjusted by sex, age, BMI, and mean blood pressure, using general linear models with the group as a fixed factor and sex, age, BMI, and mean blood pressure as covariates. The correlations were assessed with Spearman’s or Pearson’s rank correlation according to data distribution. Data distribution was assessed using Anderson-Darling and D’Agostino-Pearson tests for normal distribution. For all the analyses, a *p*-value of <0.05 was considered significant. The analyses were conducted using Prism 8.0 software (GraphPad, San Diego, CA, USA) or SPSS 29 (IBM, Armonk, NY, USA).

## Figures and Tables

**Figure 1 ijms-24-07900-f001:**
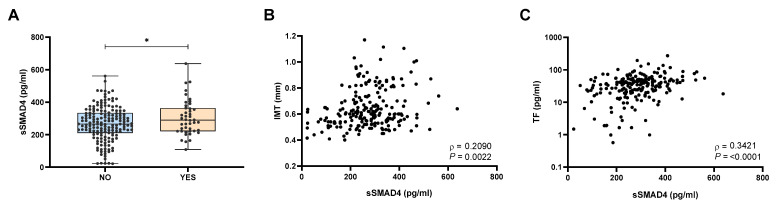
sSMAD4 is overexpressed in patients with atherosclerotic plaques. (**A**) sSMAD4 levels as determined by ELISA in plasma of subjects from the EPIOSA cohort with (n = 44) or without (n = 168) atherosclerotic plaques. Data are presented as median (interquartile range), maximum and minimum values. Comparison between groups was performed by Mann–Whitney U-test. (**B**,**C**) Correlation between sSMAD4 plasma concentration and (**B**) Carotid intima-media thickness –IMT- (n = 212) or (**C**) tissue factor (TF) levels (n = 212) in patients from the EPIOSA cohort. Spearman’s correlation coefficients (*ρ*) and *p*-values are shown. *: *p* < 0.05.

**Figure 2 ijms-24-07900-f002:**
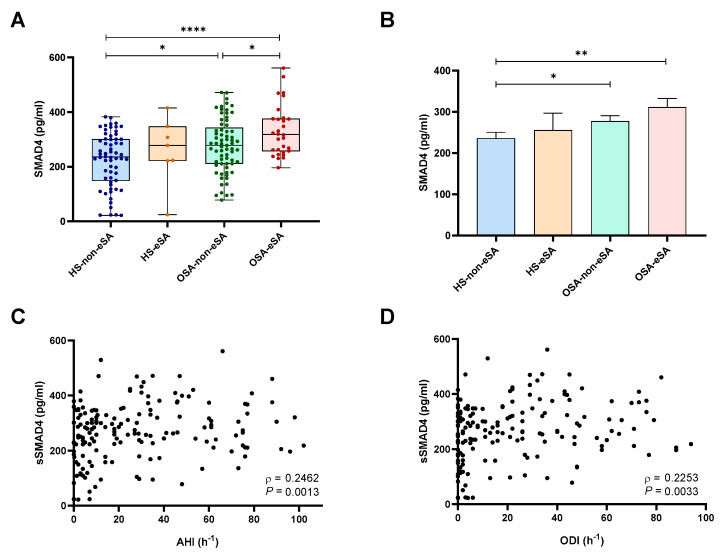
Soluble SMAD4 is overexpressed in OSA patients with early subclinical atherosclerosis. (**A**) Soluble SMAD4 (sSMAD4) levels in plasma from EPIOSA subjects: healthy subjects without early subclinical atherosclerosis (HS-non-eSA, n = 62) or with early subclinical atherosclerosis (HS-eSA, n = 7) and in OSA patients without (OSA-non-eSA, n = 69) and with early subclinical atherosclerosis (OSA-eSA, n = 30) determined by ELISA. Data are presented as median (interquartile range), maximum and minimum values. Comparisons between groups were performed by the Kruskal–Wallis test. (**B**) Comparison of sSMAD4 plasma levels adjusted by sex, age, body mass index and mean blood pressure between the study groups. The rectangles correspond to the adjusted means and the error bars to the standard error of the means. Comparisons were performed by Welch and Brown-Forsythe analysis of variance test. (**C**,**D**) Correlation between sSMAD4 plasma concentration and (**C**) apnea–hypopnea index [AHI] (n = 168) and (**D**) oxygen desaturation index [ODI] (n = 168). Spearman’s correlation coefficients (*ρ*) and *p*-values are shown. *: *p* < 0.05, **: *p* < 0.01, ****: *p* < 0.0001.

**Figure 3 ijms-24-07900-f003:**
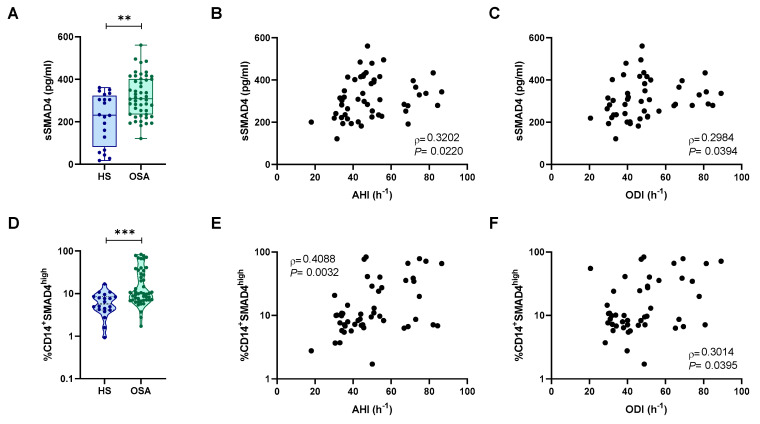
Intracellular expression of SMAD4 is increased in monocytes from OSA patients. (**A**) sSMAD4 levels in plasma from RC cohort subjects: healthy subjects (HS, n = 20) or OSA patients (OSA, n = 50) determined by ELISA. Comparison between groups was performed by the Mann–Whitney U-test. (**B**,**C**) Correlation between sSMAD4 plasma concentration and (**B**) apnea–hypopnea index [AHI] (n = 50) and (**C**) oxygen desaturation index [ODI] (n = 50). Spearman’s correlation coefficients (*ρ*) and *p*-values are shown. (**D**) SMAD4 intracellular expression in monocytes from RC cohort subjects: healthy subjects (HS, n = 20) or OSA patients (OSA, n = 50) determined by flow cytometry. Comparison between groups was performed by the Mann–Whitney U-test. (**E**,**F**) Correlation between intracellular SMAD4 expression and (**E**) apnea–hypopnea index [AHI] (n = 50) and (**F**) oxygen desaturation index [ODI] (n = 50). Spearman’s correlation coefficients (*ρ*) and *p*-values are shown. **: *p* < 0.01, ***: *p* < 0.001.

**Figure 4 ijms-24-07900-f004:**
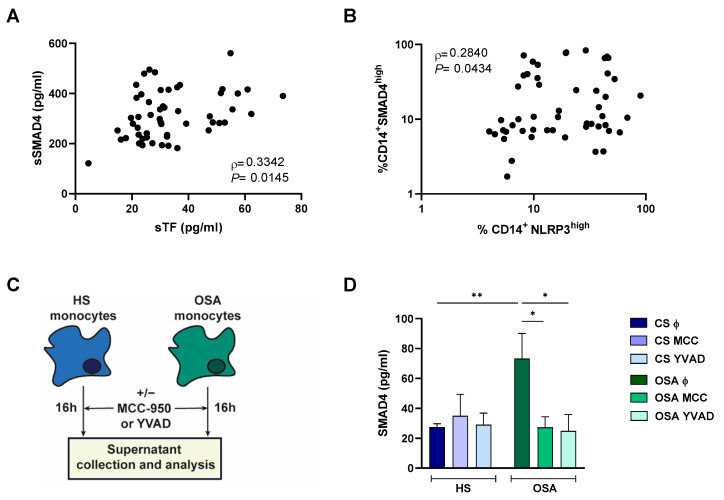
NLRP3 may mediate SMAD4 release to the plasma. (**A**) Correlation between sSMAD4 and TF plasma concentration in patients from the RC cohort (n = 50) assessed by ELISA. (**B**) Correlation between SMAD4 and NLRP3 intracellular expression in monocytes from patients from RC cohort (n = 50) assessed by flow cytometry. Spearman’s correlation coefficients (*ρ*) and *p*-values are shown. (**C**) Schematic representation of the ex vivo models, monocytes from healthy subjects (HS, n = 6) or OSA patients (OSA, n = 7) were cultured for 16 h with the presence or absence of the NLRP3 inhibitor MCC-950 (MCC) or the caspase-1 inhibitor Ac-YVAD-cmk (YVAD). (**D**) SMAD4 supernatant concentration from ex vivo models assessed by ELISA. Mean ± SEM is shown. Comparisons between groups were performed by Two-Way ANOVA with Bonferroni’s multiple comparison test. *: *p* < 0.05, **: *p* < 0.01.

**Table 1 ijms-24-07900-t001:** General characteristics of the subjects from the EPIOSA cohort.

	Subjects with Atherosclerotic Plaques	Subjects without Atherosclerotic Plaques						
	Overall	Overall	*p*-Value	Healthy Subjects		OSA Patients		*p*-Value
				Non-Early Subclinical Atherosclerosis (HS-non-eSA)	Early Subclinical Atherosclerosis (HS-eSA)	Non-Early Subclinical Atherosclerosis (OSA-non-eSA)	Early Subclinical Atherosclerosis (OSA-eSA)	
n	44	168	-	62	7	69	30	-
Male sex, n (%)	36 (81)	114 (67)	0.070	35 (56)	5 (71)	52 (75)	22 (73)	0.08
Age, years	51 (45–57)	43 (36–52)	0.001	36 (33–43)	45 (41–51) *	43 (38–58) ^#^	53 (47–59) *	<0.0001
Body mass index, kg/m^2^	30.2 (27.3–33.3)	29.5 (26.5–33.9)	0.890	27.4 (24.5–30)	30.2 (27.1–32.8)	30.1 (27.2–34.6) ^#^	32.3 (29.8–37.1)	<0.0001
Body fat, %	32.3 (28.6–37.1)	32.8 (27.2–40)	0.458	32 (23.3–40.7)	34.6 (27.7–42.1)	33.9 (27.8–39.5) ^#^	32.6 (28.6–39.6) ^#^	0.537
Current smoker, n (%)	11 (25)	37 (22)	0.675	12 (19)	0 (0)	16 (23)	9 (30)	0.334
Former smoker, n (%)	18 (41)	45 (27)	0.070	18 (29)	3 (43)	16 (23)	8 (27)	0.674
Pack-years	15 (0–30)	0 (0–14)	0.002	0 (0–14)	0 (0–19)	0 (0–11)	7 (0–27)	0.174
Epworth Sleepiness Scale	8 (5–12)	10 (6–13)	0.108	9 (6–13)	10 (8–14)	14 (2.5–33)	20 (6.7–46.2)	0.329
AHI, events/h	32.5 (14.8–61.5)	18 (5–45.6)	0.012	4 (1.2–7.2)	4 (2–8)	35.0 (21.5–62.5) ^#^	42 (27.7–59.2) ^#^	<0.0001
ODI, events/h	35.5 (15.7–51.2)	21 (2–43)	0.036	0 (0–2)	2.5 (1.5–3.5)	29 (17–49) ^#^	40 (26–60) ^#^	<0.0001
CT90, %	14 (2–35.3)	5 (0–20.5)	0.018	0 (0–2)	0 (0–1)	10 (1.5–25) ^#^	20 (6–43.5) ^#^	<0.0001
Mean nocturnal SaO_2_, %	92.5 (91–94)	94 (92–95)	0.100	95 (94–96)	94 (94–95)	93 (91–94) ^#^	92 (90–93)	<0.0001
Low nocturnal SaO_2_, %	78 (71–85)	83 (75–88)	0.018	88 (84–92)	89 (83–92)	79 (71–84) ^#^	77 (70–82) ^#^	<0.0001
Systolic BP, mm Hg	133 (128–144)	126 (116–134)	0.0002	120 (111–131)	124 (115–130)	127 (120–134)	132 (123–145)	0.0002
Diastolic BP, mm Hg	85 (76–94)	80 (71–87)	0.029	74 (68–82)	81 (73–91)	85 (76–91) ^#^	83 (75–95)	0.0001
IMT, mm	0.65 (0.58–0.76)	0.59 (0.53–0.72)	0.042	0.53 (0.50–0.57)	0.86 (0.82–0.90) *	0.59 (0.54–0.63) ^#^	0.86 (0.82–0.90) *	<0.0001

Data are expressed as number (percentage) or median (interquartile range). Comparisons between groups were performed by Mann–Whitney U-test, Kruskal–Wallis test, or chi-squared test. Abbreviations: AHI = apnea–hypopnea index; ODI = oxygen desaturation index, CT90 = Time recorded with SaO_2_ < 90%, SaO_2_ = oxyhemoglobin. IMT = intima-media thickness. * are used to indicate significance between eSA and non-eSA condition within each group. ^#^ are used to indicate significance between CS and OSA groups within the same eSA condition.

**Table 2 ijms-24-07900-t002:** General characteristics of the subjects from the RC cohort.

	Non-Apneic Healthy Subjects (n = 20)	Severe OSA Patients (n = 50)	*p*-Value
Male sex, n (%)	12 (60)	44 (73)	0.260
Age, years	51 (47–58)	58 (51–64)	0.176
Body mass index, kg/m^2^	28.3 (25.7–34.5)	31.2 (28.4–33.8)	0.105
Current smoker n (%)	4 (20)	9 (15)	0.433
Former smoker n (%)	2 (10)	11 (18.3)	0.363
Epworth Sleepiness Scale	2 (0–4)	8 (5–13)	<0.001
AHI, events/h	3.2 (1.1–5.8)	47.5 (36.4–67.25)	<0.001
ODI, events/h	3 (0–5.2)	46 (33.9–62.5)	<0.001
Mean nocturnal SaO_2_, %	94 (93–96)	90 (88–92)	<0.001
Low nocturnal SaO_2_, %	89 (82–93)	75 (65.5–80)	<0.001

Comparisons between groups were performed by Mann–Whitney U-test or chi-squared test. Abbreviations: AHI = apnea–hypopnea index; ODI = oxygen desaturation index, SaO_2_ = oxyhemoglobin saturation; BP = blood pressure.

## Data Availability

Data available on request.

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
