# Peer review of "SMAD4 Expression in Monocytes as a Potential Biomarker for Atherosclerosis Risk in Patients with Obstructive Sleep Apnea"

_ijms, 2023, doi:10.3390/ijms24097900_

Round 1
Reviewer 1 Report
Dear authors,
Thank you for submitting your manuscript entitled "SMAD4 expression in monocytes as a potential biomarker for atherosclerosis risk in patients with obstructive sleep apnea" to our journal. Overall, I found the study to be well-designed and the results to be interesting and potentially clinically relevant. However, I have a few comments and suggestions that I believe would improve the manuscript:
1. The introduction could benefit from more background information on the role of SMAD4 in atherosclerosis and cardiovascular disease. While some information is provided, it would be helpful to have a more comprehensive overview of the current state of knowledge in this area.
2.What other biomarkers were evaluated in this study besides SMAD4?
3.Have similar studies been conducted previously?
4.Were there any differences in SMAD4 expression between different types of sleep apnea (e.g., obstructive vs central)?
5.Were there any differences in SMAD4 expression between patients with mild, moderate, or severe sleep apnea?
6.Were there any differences in SMAD4 expression between patients who received treatment for sleep apnea (e.g., continuous positive airway pressure) and those who did not?
7.Were there any correlations between SMAD4 expression and other biomarkers of inflammation or oxidative stress?
8.How long did it take for changes in SMAD4 expression to occur after exposure to intermittent hypoxia, and how long did they persist after cessation
9.What are the implications of these findings for future research on sleep apnea and its complications?
10.Were there any differences in SMAD4 expression between patients with and without comorbidities such as hypertension or diabetes?
11.Were there any differences in SMAD4 expression between male and female patients?
12.Were there any differences in SMAD4 expression between different age groups?
13.Are there any potential therapeutic targets that could modulate SMAD4 expression or activity in patients with sleep apnea?
14.Were there any differences in SMAD4 expression between patients with different body mass index (BMI) categories?
15.Were there any differences in SMAD4 expression between patients who were current smokers, former smokers, or never smokers?
16. It would be helpful to include more information on potential confounding factors that could affect SMAD4 expression (e.g., medications, comorbidities) and how these were controlled for in the analysis.
17. The discussion section could benefit from how do the previous results compare to current findings?
18. The authors should provide more information on the reproducibility of their findings. For example, were the results consistent across both independent OSA cohorts? Were there any differences in SMAD4 expression between patients with different severity of OSA?
19. The conclusion could benefit from a more concise summary of the main findings and their implications for clinical practice.
Overall, I believe that this manuscript has the potential to make an important contribution to our understanding of the role of SMAD4 expression in atherosclerosis risk in patients with obstructive sleep apnea. With some revisions and additional detail, this study could be an important addition to the literature.
Sincerely
Author Response
"Please see the attachment"

Reviewer 2 Report
Diaz-Garcia and colleagues have previously demonstrated that the central regulator of transforming growth factor (TGF-β) pathway, small mother against decapentaplegic homologue 4 (SMAD4), is augmented in patients with obstructive sleep apnea (OSA). In the current paper, the same research group analyzed soluble and intracellular SMAD4 levels in plasma and monocytes from OSA patients and non-apneic subjects, with or without early subclinical atherosclerosis (eSA). Moreover, they used in vitro and ex vivo models to explore the mechanisms underlying SMAD4 upregulation and release. The researchers confirm elevated sSMAD4 levels in OSA patients and further suggest that its levels are even higher in those OSA patients with eSA. Moreover, they suggest that SMAD4 is overexpressed in monocytes from OSA patients and that intermittent hypoxia contributes to SMAD4 upregulation, which seems to be meditated by NLRP3. The authors conclude that sSMAD4 may be considered as a biomarker for atherosclerosis risk in patients with OSA. The study is well-done and the paper is well-written. I have only some minor suggestions for further improvement of the paper.
1) The title may be changed as "Intermittent hypoxia mediates SMAD4 upregulation and early subclinical atherosclerosis in patients with obstructive sleep apnea ".
2) Materials and Methods section can be presented after Introduction section.
3) You have isolated monocytes but you have not provided any single image obtained from the isolated monocytes. What was the ratio of purification in flow cytometry?
4) Clarify why AHI cut-off 10 was chosen as diagnostic criterium for OSA. The control group, no-OSA has AHI<5 events/h, and usually AHI >=15 events/h is used for moderate to severe OSA.
5) The Discussion section may have subheadings for a better readability.
6) Specify the strengths and weaknesses of the study as well as generalizability (males/females; comorbidities).
Author Response
"Please see the attachment"

Reviewer 3 Report
"Intermittent hypoxia mediates SMAD4 expression and release 2 to extracellular space in patients with sleep apnea" is a very important paper,
but
it needs some improvements:
1. linguistic style - the article has to be checked by a native speaker
2. order of content isn't correct
3. references has to be edited (e.g. Nr. 39).
Author Response
- linguistic style - the article has to be checked by a native speaker
R1. Thank you for your suggestion. We have improved the language of our manuscript, which has been revised by a native speaker.
- order of content isn't correct
R2. Apologize the inconvenience. We have follow the MDPI template recommended from the official journal website.
- references has to be edited (e.g. Nr. 39).
R3. Apologize the inconvenience. We have corrected reference 39 and others (please notice that reference 39 has been performed by a high number of authors and we are not able to omit some of them due to journal rules).
Round 2
Reviewer 1 Report
Good revisions. I have no further questions.
Reviewer 3 Report
"Intermittent hypoxia mediates SMAD4 expression and release to extracellular space in patients with sleep apnea" can be published.
The manuscript improved after revision: The authors include the advices of the revierwers.